# Study of Magnetic Properties and Relaxation Time of Nanoparticle Fe_3_O_4_-SiO_2_

**DOI:** 10.3390/ma15041573

**Published:** 2022-02-19

**Authors:** Togar Saragi, Bayu Permana, Arnold Therigan, Hotmas D. Sinaga, Trisna Maulana, Risdiana Risdiana

**Affiliations:** Department of Physics, Faculty of Mathematics and Natural Sciences, Universitas Padjadjaran, Jl. Raya Bandung-Sumedang km 21, Jatinangor, Sumedang 45363, Indonesia; bayubaper@gmail.com (B.P.); arnoldtherigan15@gmail.com (A.T.); hotmas28@gmail.com (H.D.S.); trisna17002@mail.unpad.ac.id (T.M.); risdiana@phys.unpad.ac.id (R.R.)

**Keywords:** encapsulation, Fe_3_O_4_ nanoparticle, irreversibility, relaxation time, SiO_2_, superparamagnetic

## Abstract

The magnetic properties and relaxation time of Fe_3_O_4_ nanoparticles, and their encapsulation with silicon dioxide (Fe_3_O_4_-SiO_2_), have been successfully investigated by analyzing the temperature dependence of magnetization (M(T)) and the time dependence of magnetization (M(t)), using the SQUID magnetometer measurement. The M(T) measurement results can determine the magnetic parameters and magnetic irreversibility of Fe_3_O_4_ and Fe_3_O_4_-SiO_2_ samples. The values of Curie constant (C), effective magnetic moment ($\mu_{eff}$), and Weiss temperature ($\theta_{P}$) are 4.2 (emu.K.Oe/mol), 5.77 $\mu_{B}$, and −349 K, respectively, for the Fe_3_O_4_ samples, and 81.3 (emu.K.Oe/mol), 25.49 $\mu_{B}$, and −2440 K, respectively, for the Fe_3_O_4_-SiO_2_ samples. After encapsulation, the broadening peak deviation decreased from 281.6 K to 279 K, indicating that the superparamagnetic interactions increased with the encapsulation process. The magnetic parameters and irreversibility values showed that the superparamagnetic properties increased significantly after encapsulation (Fe_3_O_4_-SiO_2_). From the results of the M(t) measurement, it was found that there was a decrease in the magnetic relaxation time after the encapsulation process, which indicated that the distribution of the nanoparticle size and anisotropy energy increased.

## 1. Introduction

Fe-based materials are magnetic materials that have many functions and applications that can be developed. The most studied Fe-based materials are Fe_2_O_3_ and Fe_3_O_4_. These two materials have different characteristics that can be utilized for specific applications. In the bulk state, the Fe_2_O_3_ material has antiferromagnetic and weak ferromagnetism properties in a certain temperature range, while the Fe_3_O_4_ material has ferrimagnetic properties in the entire temperature range [1]. Fe_3_O_4_ is a material that has widely attracted attention to be studied, especially for several applications, such as in ferrofluids [2], magnetic refrigeration, the detoxification of biological fluids, and the magnetically controlled transport of anticancer drugs [3,4]. This material is still open and exciting to study, especially at the nano size. At the nanoparticle size, Fe_3_O_4_ has superparamagnetic properties, where the magnetization can fluctuate thermally. The magnetic saturation is high, and the coercivity and remanence are equal to zero. Therefore, this nanoparticle can be delivered to the tissue target without agglomeration, and can easily be controlled by a magnetic field [5]. Fe_3_O_4_ is quite a stable compound compared to Fe_2_O_3_, so it is very suitable for observing various magnetic properties of the material.

The superparamagnetic state can be realized through encapsulation processes, such as SiO_2_ [3,4] and oleic acid [6]. This process aims to prevent agglomeration, maintain the magnetic stability of the material, and reduce the cytotoxic effect. In this study, SiO_2_ was chosen as a ligand to modify the surface of the Fe_3_O_4_ nanoparticles. SiO_2_ is a compound that has good chemical stability, good hydrophilicity, and has a surface hydroxyl group that allows further modification. In addition, SiO_2_ is not easy to agglomerate during the synthesis process, undergoes intermolecular redox reactions, and can reduce the cytotoxic effect of the Fe_3_O_4_ nanoparticles when applied for various biomedical purposes [4].

The magnetic properties of nanoparticles, both Fe_3_O_4_ and Fe_3_O_4_-SiO_2__,_ are very important to study. These magnetic characteristics can provide information about important parameters that can be considered in their development and utilization for various applications. One of these characteristics is the temperature dependence of magnetization M(T) and the time dependence of magnetization M(t). The information obtained from the M(T) measurement includes the type of magnetic material, the blocking temperature, the critical temperature, and the effective magnetic moment [1,7,8,9], while the M(t) measurement obtained relaxation time information. The relaxation time is a measure of the rotational freedom of magnetic nanoparticles, which provides information on several material characteristics, such as the viscosity, chemical bonding, and stiffness of the matrix to which the nanoparticles are bonded. This characteristic is an important parameter for biological applications [10].

In previous research, we have analyzed M(T) data on the ZFC process for Fe_3_O_4_ and Fe_3_O_4_-SiO_2_ nanoparticles, to obtain information on the effect of SiO_2_ encapsulation on the blocking temperature of the Fe_3_O_4_ material. We found that the blocking temperature increased after the encapsulation process [8]. The value of $T_{C}$ has also been reported for Fe_3_O_4_-SiO_2,_ by measuring the samples in the temperature range of 300–900 K and a magnetic field of 10 kOe. It was found that $T_{C}$ is ~850 K [11]. Therefore, the value of $T_{C}$ and other magnetic properties in the temperature range of 10–300 K still need to be determined. In addition, the effect of SiO_2_ encapsulation to Fe_3_O_4_ on the effective magnetic moment and relaxation time has not been widely reported. This information is beneficial for its development and application in biomedicine, which is difficult to obtain from in vivo research.

Here, we reported the magnetic properties of the Fe_3_O_4_ and Fe_3_O_4_-SiO_2_ nanoparticles, namely, the Curie constant (C), effective magnetic moment $\left(\mu_{eff}\right)$, Weiss temperature ($\theta_{P}$) from the M(T) measurement in the FC process, and the relaxation time (τ) from the M(t) measurement [12]. We also analyzed the magnetothermal properties by plotting [$-d\left(M_{FC}-M_{ZFC}\right)/dT$] versus temperature. This analyst will provide information regarding peak broadening, which describes the distribution of nanoparticles, concerning the anisotropic energy of the system [13,14,15].

## 2. Materials and Methods

Magnetic nanoparticles of Fe_3_O_4_ were synthesized by the co-precipitation method. The precursors of ${\mathrm{Fe}}^{2+}$ and ${\mathrm{Fe}}^{3+}$ cations are ferric chlorid hexahydrate, FeCl_3_·6H_2_O (CAS No. 10025-77-1, Sigma-Aldrich, St. Louis, MI, USA), and iron (II) chloride tetrahydrate puriss. p.a, FeCl_2_·4H_2_O (CAS No. 13478-10-9, Sigma-Aldrich, St. Louis, MI, USA), respectively. A total of 5.41 g of FeCl_3_·6H_2_O and 1.99 g of FeCl_2_·4H_2_O were dissolved in 100 mL of DI water, then stirred for 30 min at 25 °C. Experiment details are described in our previous publications [16].

### 2.1. Synthesis of Fe_3_O_4_ Nanoparticles

From 100 mL of the solution resulting from mixing FeCl_3_·6H_2_O and FeCl_2_·4H_2_O precursors, 25 mL of the solution was separated, and 25% NH_4_OH was added dropwise until pH = 10. Then the solution was stirred and sonicated for 30 min. In this study, the sonication process was carried out simply by using a sonication bath (Branson M1800-E, CPX-952-136R, Shanghai, China) without a sonicator tip. The sonication bath was filled with water, and a beaker glass containing the magnetite solution was placed into the sonication bath. The frequency used in the sonication bath was 40 kHz. This ultrasonic wave was delivered to a glass beaker containing a magnetite solution through water in the sonication bath. After sonication, the solution was allowed to stand for 30 min to obtain a precipitate of magnetite nanoparticles. The precipitate formed was washed with n-hexane and re-dispersed into DI water. The sample obtained is referred to as Fe_3_O_4_.

### 2.2. Synthesis of Fe_3_O_4_-SiO_2_ Nanoparticles

From 75 mL of the remaining solution resulting from mixing FeCl_3_·6H_2_O and FeCl_2_·4H_2_O precursors, 25 mL of the solution was separated again, and 0.4 mL of tetraethyl ortho-silicate (TEOS) was added, and, finally, 25% NH_4_OH was added dropwise until pH = 10. The same process was carried out as the synthesis process of Fe_3_O_4_. The sample is referred to as Fe_3_O_4_-SiO_2_.

### 2.3. Characterization

All samples were characterized by using high-resolution transmission electron microscopy (HR-TEM H9500, Hitachi High-Tech, Tokyo, Japan) to measure the particle size. The temperature dependence of magnetization M(T) in zero fields cooled (ZFC) and field cooled (FC), with the external magnetic field of 100 Oe at 10–300 K, was measured by a superconducting quantum interference device (SQUID) Magnetometer (Quantum Design MPMS XL, San Diego, CA, USA). We also measured the time dependence of magnetization M(t) to investigate the relaxation time of magnetite Fe_3_O_4_ and Fe_3_O_4_-SiO_2_. The SQUID measurements were carried out in the Graduate School of Engineering, Tohoku University, Japan.

## 3. Results and Discussion

### 3.1. Particle Size of Fe_3_O_4_ and Fe_3_O_4_-SiO_2_ Nanoparticles

Figure 1 shows HR-TEM images of the Fe_3_O_4_ (a) and Fe_3_O_4_-SiO_2_ (b) samples. In general, the morphology of the two samples was nearly spherical. The Fe_3_O_4_ sample shows a clear boundary, while the Fe_3_O_4_-SiO_2_ sample shows a shadow boundary, which is believed to be the result of the encapsulation process. It can also be observed that the particle size of the Fe_3_O_4_ sample is 11 ± 2.22 nm, while the particle size of the core Fe_3_O_4_-SiO_2_ sample is 10 ± 1.53 nm.

### 3.2. Temperature Dependence of Magnetization, M(T)

The results of the temperature dependence of the magnetization of M(T)_FC_ and M(T)_ZFC_ for the Fe_3_O_4_ (a) and Fe_3_O_4_-SiO_2_ (b) nanoparticles are shown in Figure 2.

From Figure 2a, it can be observed that the magnetization value of Fe_3_O_4_ slowly increases when the temperature is lowered to about 100 K, for both ZFC and FC. At temperatures below 100 K in the ZFC process, the magnetization value decreases sharply, while, in the FC process, the magnetization value tends to be independent of temperature below about 80 K. From Figure 1b, it can be observed that changes in the magnetization value of Fe_3_O_4_-SiO_2_ occur at a higher temperature range. The value of magnetization in the ZFC process, after encapsulation, increased when the temperature decreased from 300 K to 217 K, and then decreased when the temperature decreased to 10 K. In the FC process, the magnetization value tends to increase slightly as the temperature decreases from 300 K to 10 K.

Figure 2 also shows the characteristics of magnetic irreversibility, which reveals differences in the magnetization values of $M_{FC}$ and $M_{ZFC}$ in the entire measurement temperature range. It indicates a thermally induced magnetic relaxation process. The effect of magnetic irreversibility is shown in Fe_3_O_4_ and Fe_3_O_4_-SiO_2_, where the $M_{FC}$ and $M_{ZFC}$ branches meet at 300 K. Although these branches do not overlap below 300 K, it can be observed that Fe_3_O_4_ and Fe_3_O_4_-SiO_2_ exhibit superparamagnetic characteristics. The branching point temperature of the $M_{FC}$ and $M_{ZFC}$ in the Fe_3_O_4_-SiO_2_ sample is lower than that in the Fe_3_O_4_ sample, namely, at temperatures around 261 K and 300 K, respectively. This is in accordance with the $H_{C}$ coercivity value for the Fe_3_O_4_-SiO_2_ sample, which is lower than that of the Fe_3_O_4_ samples, which are 25.99 Oe and 362.37 Oe, respectively [16].

### 3.3. Curie Temperature (T_C_)

This change in the magnetization tendency to temperature can be further analyzed to determine the Curie temperature. Figure 3 shows the magnetization derivative curve of M(T)_FC_ to the temperature (dM/dT) of Fe_3_O_4_. From the curve (dM/dT) versus T at the minimum peak, we can estimate the Curie temperature ($T_{C}$) of the sample [9,11]. From Figure 3, it can be estimated that the $T_{C}$ value for Fe_3_O_4_ is 125 K. This value is relatively smaller than that reported by Blundell, which is 858 K [17], probably due to differences in particle size or cation vacancies. As reported in previous studies, the $T_{C}$ value increases with decreasing particle size [18] or increasing cation vacancies [11]. On the other hand, the Curie temperature of the encapsulated Fe_3_O_4_ with SiO_2_ could not be determined. This is because there is no significant change in magnetization ($M_{FC}$) when the temperature is lowered, as shown in Figure 2b, so the minimum peak of the first derivative (dM/dT) is not obtained.

### 3.4. Curie Constant, Effective Magnetic Moment, and Weiss Temperature

The values of C, $\mu_{eff}$, and $\theta_{P}$ can be determined by using the Curie–Weiss law in Equations (1) and (2). The value of the Curie constant is obtained through the following regression equation (Equation (1)) of the inverse molar susceptibility curve ($\chi^{-1}$) versus temperature (T):(1)$$\chi_{m}=\frac{C}{\left(T-\theta_{P}\right)}\;$$
(2)$$C=\frac{N_{A}}{3k_{B}}\mu_{eff}{}^{2}\mu_{B}{}^{2}$$
where C is the Curie constant, defined as Equation (2), $\theta_{P}$ is the Weiss temperature, $N_{A}$ is Avogadro’s number, $k_{B}$ is the Boltzmann constant, and $\mu_{B}$ is the Bohr magneton. From Equation (2), the value of the effective magnetic moment is obtained ($\mu_{eff}=\sqrt{3Ck_{B}/N_{A}\mu_{B}{}^{2}}=2.828\sqrt{C}$) [9,19]. From Equation (1), the Curie constant and the Weiss temperature can be obtained through the regression equation ($\chi^{-1}$) versus temperature (T), as shown in Figure 4.

By using the C value from the fitting and Equation (2), the effective magnetic moment value is obtained. These magnetic quantities are shown in Table 1. For reference, we also displayed the $H_{C}$ values for the Fe_3_O_4_ and Fe_3_O_4_-SiO_2_ samples [16].

From Table 1, it can be observed that the effective magnetic moment of ${\mathrm{Fe}}^{3+}$ in the Fe_3_O_4_ sample is 5.77 $\mu_{B}$. This value is very compatible with the experimental and theoretical results reported by Zatsiupa et al., for the case of Bi_25_FeO_39_ at a temperature of 5–950 K and a magnetic field of 0.86 T [19]. From these experimental and theoretical results, the values of the effective magnetic moments of the ${\mathrm{Fe}}^{3+}$ ion are reported as 5.82 $\mu_{B}$ and 5.92 $\mu_{B}$, respectively [19]. The value of the effective magnetic moment of the sample after encapsulation (Fe_3_O_4_-SiO_2_) increased four times, from 5.77 $\mu_{B}$ to 25.49 $\mu_{B}$. This increment is probably due to the emergence of superparamagnetism from single-domain nanomagnets. In one unit cell of the FCC Fe_3_O_4_ or (${\mathrm{Fe}}^{2+}$)[${\mathrm{Fe}}_{2}^{3+}$]$O_{4}$ system, there are eight bivalent ${\mathrm{Fe}}^{2+}$ cations in the tetrahedral site, which are antiparallel to four cations. Another trivalent is ${\mathrm{Fe}}^{3+}$, so the effective moment of Fe_3_O_4_-SiO_2_ is four times higher than that of Fe_3_O_4_ before encapsulation [17]. The effective magnetic moment of a nanoparticle increases with a decrease in the particle size. Increasing the value of $\mu_{eff}$ by four times is expected to reduce the value of the magnetic force by up to four times [1], to eliminate the suspected target, if Fe_3_O_4_-SiO_2_ was applied as the material for magnetic hyperthermia applications.

### 3.5. Distribution of the Anisotropy Energy Barriers

The distribution of the anisotropic energy barriers of the system, which is closely related to the distribution of nanoparticle size and blocking temperature [13,14,15], can be analyzed through the $M_{FC}$ and $M_{ZFC}$ derivative curves to temperature, [$-d\left(M_{FC}-M_{ZFC}\right)/dT$]. Figure 5 shows the derivative curve of [$-d\left(M_{FC}-M_{ZFC}\right)/dT$] versus the temperature of the Fe_3_O_4_ and Fe_3_O_4_-SiO_2_ samples.

From Figure 5, it can be observed that the Fe_3_O_4_ sample has one broadening peak at a temperature of 281 K, while the Fe_3_O_4_-SiO_2_ sample has a broadening peak at two temperatures, namely, 51.4 K and 279 K. At high temperatures, the broadening peak after encapsulation shifts to lower temperatures, from 281.6 K to 279 K. This indicates that there is dependence of the energy distribution of the barrier on the encapsulation process, which is also related to the nanoparticle size distribution [14]. On the other hand, the shift also indicates relaxation of the almost free assembly of particles, due to superparamagnetism interactions. At low temperatures, the broadening peak of the Fe_3_O_4_ samples was not observed, probably due to the limitation of the measurement range. We hypothesize that the peak temperature broadening of the Fe_3_O_4_ sample in the temperature range below 51.4 K might be observed if we conducted the measurement in a narrower temperature range. From our previous study, it has been reported that the blocking temperature and anisotropy energy (∆E) of Fe_3_O_4_ and its encapsulated Fe_3_O_4_-SiO_2_ are 118.38 K and 209.03 K, and $3.0\times 10^{3}\;k_{B}$ and 5$.2\times 10^{3}\;k_{B}$, respectively [8]. An increase in the blocking temperature value, caused by the encapsulation process, indicates an increase in magnetic interactions between the nanoparticles, resulting in an increase in anisotropic energy barriers, causing the reversal of the magnetization. This is similar to the results obtained by Del Bianco et al., for the case of Fe_3_O_4_, with various applied fields [14].

### 3.6. Magnetic Relaxation Time, M(t)

Figure 6 shows the magnetization versus time curve after the external magnetic field is quenched to zero, and following curve fitting. Figure 6a,c, show the magnetization versus time curve for Fe_3_O_4_ at 10 K and 300 K, while Figure 6e shows the magnetization versus time curve for Fe_3_O_4_-SiO_2_ at 10 K. Figure 6b,d show the curve fittings, using Equation (3), for Fe_3_O_4_ at 10 K and 300 K, while Figure 6f shows the fitting curve, using Equation (3), for a sample of Fe_3_O_4_-SiO_2_ at 10 K. The measurement results show that the magnetic particles relax when the external magnetic field is quenched. The characteristics of relaxation time can be analyzed using Equation (3), as follows:(3)$$M\left(t\right)=M\left(0\right)\exp\left(\frac{t}{\tau }\right)\;$$

The results of the calculation of the relaxation time value for each measurement are arranged in Table 2. The magnetic relaxation times of Fe_3_O_4_ at 10 K and 300 K are 7.10 × 10^3^ s and 5.62 × 10^3^ s, respectively, while the relaxation time of Fe_3_O_4_-SiO_2,_ after encapsulation, decreased to 3.57 × 10^3^ s. In general, the relaxation time of the Fe_3_O_4_ nanoparticles is greater than that of the Fe_3_O_4_-SiO_2_ nanoparticles after encapsulation. The lower relaxation times are associated with better superparamagnetic properties. These better superparamagnetic properties are supported by the $H_{C}$ value of the Fe_3_O_4_-SiO_2_ sample ($H_{C}$ = 25.99 Oe), which is much lower than the $H_{C}$ value of the Fe_3_O_4_ sample ($H_{C}$ = 362.37 Oe) [16]. The lower relaxation time is also associated with an increase in the concentration of nanoparticles in Fe_3_O_4_-SiO_2_ [7]. The decrease in relaxation time in Fe_3_O_4_, after being encapsulated with SiO_2,_ can also be related to changes in the anisotropic energy value of the nanoparticles, where the anisotropic energy is also related to the blocking temperature [8,14,20,21]. In our previous study, it was reported that the anisotropic energy of the nanoparticles increased when the Fe_3_O_4_ samples were encapsulated with SiO_2_. The anisotropic energy of the Fe_3_O_4_ nanoparticles is $3.00\times 10^{3}\;k_{B}$, while for the Fe_3_O_4_-SiO_2_ nanoparticles, it is $5.20\times 10^{3}\;k_{B}$ [8]. It is clarified that the value of relaxation time decreases when the value of anisotropic energy is increased. The decrease in relaxation time after encapsulation can be correlated with the decrease in medium viscosity, which becomes potential information for biomedical applications [22].

In order to obtain more comprehensive information, it is necessary to conduct further research on the magnetic properties and relaxation time, depending on the size of the core Fe_3_O_4_. In addition, it is also necessary to study the relaxation time depending on the magnetic field. This study is very important to provide information about material properties for biomedical applications that cannot be obtained from in vivo studies.

## 4. Conclusions

The magnetic properties of Fe_3_O_4_ and its encapsulation with SiO_2_ (Fe_3_O_4_-SiO_2_) have been successfully investigated by analyzing the M(T) and M(t) curves of the SQUID magnetometer measurements. The M(T) curve for the Fe_3_O_4_ and Fe_3_O_4_-SiO_2_ materials show superparamagnetic properties with magnetic irreversibility characteristics. After encapsulation, the broadening peak of the first derivative of the difference between field-cooled and zero-field-cooled magnetization shifted from 281.6 K to 279 K, which indicates that the Fe_3_O_4_-SiO_2_ material has better superparamagnetic properties than the Fe_3_O_4_ sample. The values of C, $\mu_{eff}$, and $\theta_{P}$ increased significantly after encapsulation, which also supported the enhancement of the superparamagnetic properties of Fe_3_O_4_-SiO_2_. The value of the effective magnetic moment after encapsulation increased four times, from 5.77 $\mu_{B}$ to 25.49 $\mu_{B}$. This increment is probably due to the emergence of superparamagnetism from single-domain nanomagnets. Increasing the value of $\mu_{eff}$ by four times is expected to reduce the value of the magnetic force to eliminate the suspected target in magnetic hyperthermia applications.

The M(t) curve for Fe_3_O_4_ and Fe_3_O_4_-SiO_2_ shows that the magnetic relaxation time of the Fe_3_O_4_-SiO_2_ sample decreases compared to the magnetic relaxation time of Fe_3_O_4_. The lower relaxation times are associated with better superparamagnetic properties. This also indicates that the superparamagnetic properties increase with the encapsulation process. The decrease in relaxation time after encapsulation, from $7.10\times 10^{3}\;s$ to 3.57$\times 10^{3}\;s,$ can be correlated with the decrease in medium viscosity, which becomes potential information for biomedical applications.

In order to obtain more comprehensive information, it is necessary to conduct further research on the magnetic properties and relaxation time, depending on the size of the core Fe_3_O_4,_ to provide information about material properties for biomedical applications that cannot be obtained from in vivo studies.

## Figures and Tables

**Figure 1 materials-15-01573-f001:**
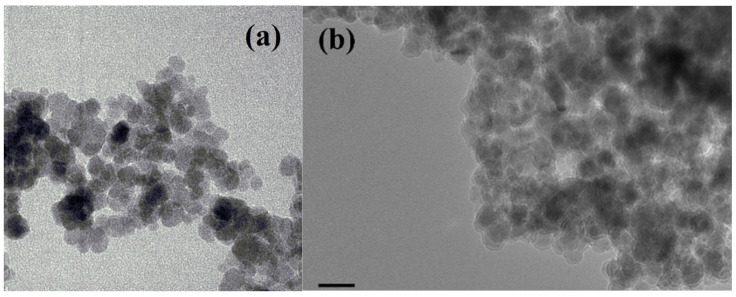
HR-TEM images of sample Fe_3_O_4_ (**a**) and Fe_3_O_4_-SiO_2_ (**b**).

**Figure 2 materials-15-01573-f002:**
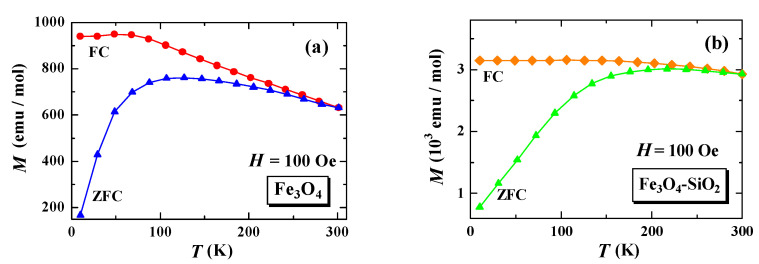
Temperature dependence of magnetization of Fe_3_O_4_ (**a**) and Fe_3_O_4_-SiO_2_ (**b**) nanoparticles.

**Figure 3 materials-15-01573-f003:**
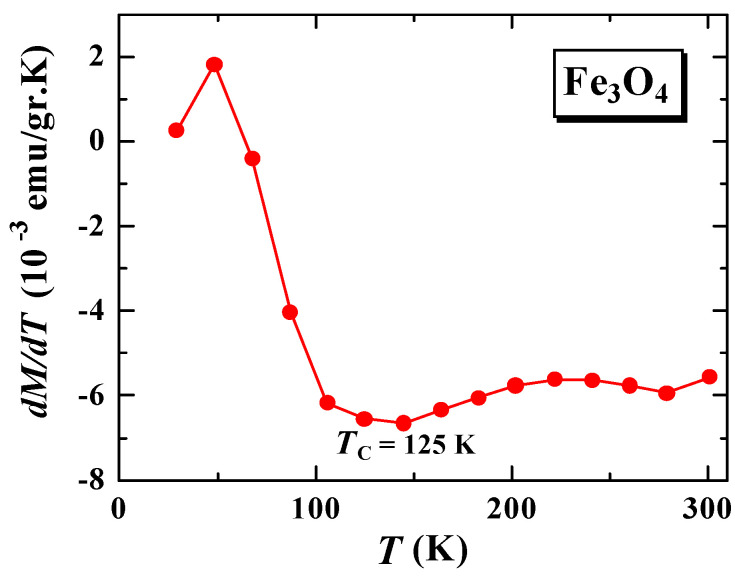
The first derivative curve of M(T)_FC_ versus temperature of sample Fe_3_O_4_.

**Figure 4 materials-15-01573-f004:**
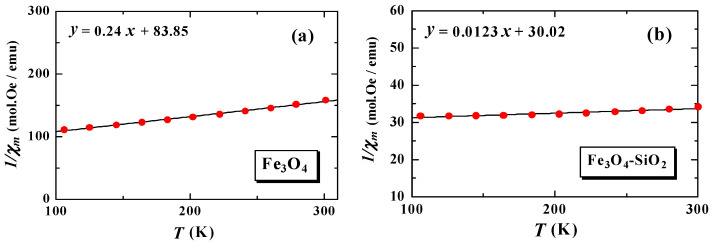
The fitting of Curie–Weiss law on $M_{FC}$ data of Fe_3_O_4_ (**a**) and Fe_3_O_4_-SiO_2_ (**b**).

**Figure 5 materials-15-01573-f005:**
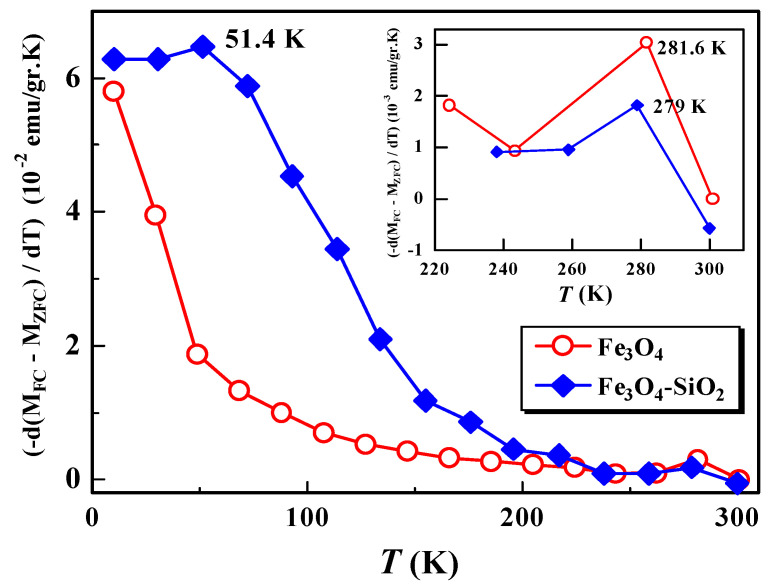
Temperature derivative [$-d\left(M_{FC}-M_{ZFC}\right)/dT$] of the difference between FC and ZFC magnetizations for samples Fe_3_O_4_ and Fe_3_O_4_-SiO_2_.

**Figure 6 materials-15-01573-f006:**
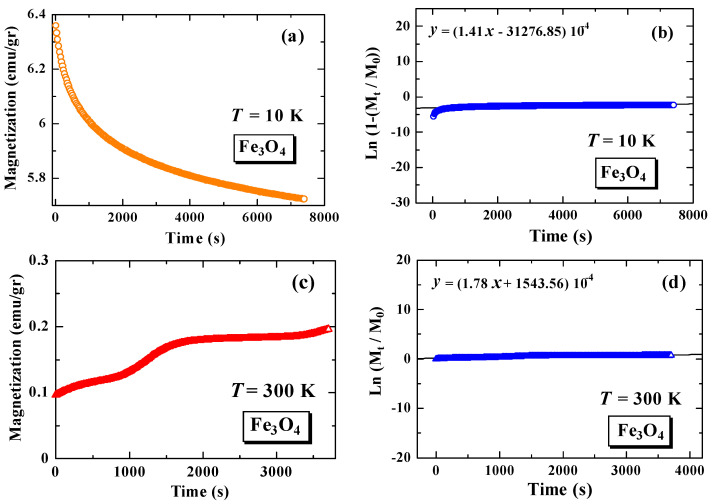
Time dependence of magnetization for Fe_3_O_4_ at *T* = 10 K (**a**), fitting curve of Fe_3_O_4_ at *T* = 10 K (**b**), Fe_3_O_4_ at *T* = 300 K (**c**), fitting curve of Fe_3_O_4_ at *T* = 300 K (**d**), Fe_3_O_4_-SiO_2_ at *T* = 10 K (**e**), and fitting curve of Fe_3_O_4_-SiO_2_ at *T* = 10 K (**f**).

**Table 1 materials-15-01573-t001:** The magnetic quantities of Fe_3_O_4_ and Fe_3_O_4_-SiO_2_.

Sample	C (emu.K.Oe/mol)	$\mu_{eff}\;\left(\mu_{B}/fu\right)$	$\theta_{P}\;(K)$	$H_{C}\;(\mathrm{Oe})$ [16]
Fe_3_O_4_	4.2 ± 0.01	5.77 ± 0.11	−349 ± 1.03	362.37
Fe_3_O_4_-SiO_2_	81.3 ± 4.20	25.49 ± 2.05	−2440 ± 126.05	25.99

**Table 2 materials-15-01573-t002:** The magnetic relaxation time of Fe_3_O_4_ and Fe_3_O_4_-SiO_2_ nanoparticles.

Sample	*T* (K)	$\tau \times 10^{3}\left(s\right)$
Fe_3_O_4_	10	7.10 ± 1.83
Fe_3_O_4_	300	5.62 ± 0.49
Fe_3_O_4_-SiO_2_	10	3.57 ± 0.67

## Data Availability

Not applicable.

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
