# Peer review of "Study of Magnetic Properties and Relaxation Time of Nanoparticle Fe3O4-SiO2"

_materials, 2022, doi:10.3390/ma15041573_

Round 1

Reviewer 1 Report

The magnetic properties and relaxation time of Fe3O4 nanoparticles and their encapsulation with silicon dioxide (Fe3O4.SiO2) have been successfully investigated by analyzing the temperature dependent of magnetization (?(?)) and the time dependent of magnetization (?(?)) using the 12 SQUID magnetometer measurement. Overall, the article is well organized and its presentation is good. However, some minor issues still need to be improved before publication. (1) The particle size of 109 the Fe3O4 sample is 11± 2.22 nm, and the particle size of the core Fe3O4.SiO2 sample is 10 110 ±1.53 nm as shown in Figure1 (Page3, Line109-111), could the author try to explain the reason of the different particle size for Fe3O4 and Fe3O4.SiO2? (2) Page5, Line 175, missing reference after the description “This value is very compatible with the experimental and theoretical 174 results reported by Zatsiupa et al.,”. (3) The format of secondary headings should be uniform. In the text, 2.1, 2.2, 2.3, 3.3 and 3.5 have no full stops. While for 3.1, 3.2, there is a full stop at the end of secondary heading. Please check. (4) Page5, Line 178-179, “The value of the effective magnetic moment of the sample after encapsulation 178 (Fe3O4-SiO2) increased four times from 5.77 ?? to 25.49 ??.”, could the author give some examples of specific applications that could benefit from the increase of the effective magnetic moment. (5) What’s the potential application for the different magnetic relaxation times for Fe3O4 (7.10×103 s) and Fe3O4-SiO2 (3.57×103 s). (6) The limitation of this work is suggested to be discussed.

Author Response

Respond to Reviewer (1)

Comments and Suggestions for Authors

The magnetic properties and relaxation time of Fe3O4 nanoparticles and their encapsulation with silicon dioxide (Fe3O4.SiO2) have been successfully investigated by analyzing the temperature dependent of magnetization (?(?)) and the time dependent of magnetization (?(?)) using the 12 SQUID magnetometer measurement. Overall, the article is well organized and its presentation is good. However, some minor issues still need to be improved before publication.

Respond: Authors would like to thank for valuable comments and suggestions to increase the quality of our manuscript.

 Reviewer (1):

  1. The particle size of the Fe3O4 sample is 11± 2.22 nm, and the particle size of the core Fe3O4.SiO2 sample is 10±1.53 nm as shown in Figure1 (Page3, Line109-111), could the author try to explain the reason of the different particle size for Fe3O4 and Fe3O4.SiO2?

Respond (1):

In our opinion, the encapsulation of Fe3O4 with SiO2 causes a change in the size of the Fe3O4 core to 10 nm. The encapsulation process consists of: before adding SiO2, the solution containing Fe3O4 nanoparticles is stirred again with a magnetic stirrer at a temperature of 80 °C. During this process, we believe that the particle size will become smaller, and when SiO2 is added, the nanoparticles are encapsulated by SiO2. Therefore, the core size of Fe3O4 is maintained at 10 nm.

 Reviewer (2)

  1. Page5, Line 175, missing reference after the description “This value is very compatible with the experimental and theoretical results reported by Zatsiupa et al.,”.

Respond (2):

We add the references [19] to support the description in page 6, line 191 “This value is very compatible with the experimental and theoretical results reported by Zatsiupa et al., for the case of Bi25FeO39 at a temperature of 5-950 K and a magnetic field of 0.86 T [19]”.

Reviewer (3)

  1. The format of secondary headings should be uniform. In the text, 2.1, 2.2, 2.3, 3.3 and 3.5 have no full stops. While for 3.1, 3.2, there is a full stop at the end of secondary heading. Please check.

Respond (3):

We have revised according to the reviewer's suggestion and also follow the Microsoft Word template for “Materials” (without full stops).

Reviewer (4)

Page5, Line 178-179, “The value of the effective magnetic moment of the sample after encapsulation 178 (Fe3OSiO2) increased four times from 5.77 to 25.49 .”, could the author give some examples of specific applications that could benefit from the increase of the effective magnetic moment.

Respond (4):

In a rough approximation, the effective magnetic moment can estimate the saturation field value by the equation , where Hs is the saturation field (magnetic force). Thus, the value of Hs is inversely proportional to the effective magnetic moment. Increasing the value of  by 4 times is expected to reduce the value of the magnetic force up to four times. This magnetic force can be used in biomedical applications such as magnetic separation, drug delivery, and magnetic hyperthermia. Reducing the value of Hs can be advantageous because in its application only a small magnetic force can generate sufficient heat to eliminate the suspected target.

We also added the explanation in page 6 Line 200-202: Increasing the value of  by 4 times is expected to reduce the value of the magnetic force up to four times [1] to eliminate the suspected target if Fe3O4.SiO2 was applied as a material for magnetic hyperthermia application.

Reviewer (5)

  1. What’s the potential application for the different magnetic relaxation times for Fe3O4 (7.10×103 s) and Fe3O4-SiO2 (3.57´103 s).

Respond (5):

The relaxation time depends on non-magnetic parameters such as the viscosity,h of the medium and the hydrodynamic volume, VH, by equation:

In this report, it is found that the relaxation time decrease after encapsulation. In biomedical applications, the value of relaxation time has a potential information for paramedics to administer drugs according to the patient's blood viscosity. For example, the lower the relaxation value of Fe3O4.SiO2 (3.57´103 s), the patient receiving this biomedical material must have a low blood viscosity as well.

We added the explanation in page 8 Line 261-263: The decreasing of relaxation time after encapsulation can be correlated with the de-crease of medium viscosity, which become potential information for biomedical applications [22].

Reviewer (6)

  1. The limitation of this work is suggested to be discussed.

Respond (6):

We have added an explanation of limitations and future work in the discussion section as follows:

Page 8 Line 266-270: In order to obtain more comprehensive information, it is necessary to conduct further research on the magnetic properties and relaxation time depending on the size of the core Fe3O4. In addition, it is also necessary to be studied the relaxation time depending on the magnetic field. This study is very important to provide information about material properties for biomedical applications that cannot be obtained from in-vivo studies.

List of Changes:

  1. Page 2 Line 50, “The magnetic characteristics” has been changed to be “the magnetic properties”.
  2. We have improved the “Introduction”. We have added the gap of research that needs to be filled displaying in page 2 Line 66-71: “Therefore, the determination of the value of and other magnetic properties in the temperature range of 10-300 K is still needed to be investigated. In addition, the effect of SiO2 encapsulation to Fe3O4 on the effective magnetic moment and relaxation time has not been widely reported. This information is beneficial for its development and application in biomedicine, which is difficult to obtain within in-vivo research.”
  3. Page 2 Line 81-83, we added the explanation about specification of precursor: The precursors of Fe2+ and Fe3+ cations are ferric chlorid hexahydrate, FeCl3.6H2O (Hoshi Rikagakukikai #15-1140-5-25G-J >99%), and iron (II) chloride tetrahydrate puriss. p.a, FeCl2.4H2O (Hoshi Rikagakukikai # 44939-5G >99%), respectively.
  4. Page 2 Line 90-95: Then the solution was stirred and sonicated at room temperature for 30 minutes. In this study, the sonication process was carried out simply by using a sonication bath without a sonicator tip. The sonication bath was filled with water, and a beaker glass containing the magnetite solution was put into the sonication bath. The frequency used in the sonication bath is 40 kHz. This ultrasonic wave will be forwarded to a glass beaker containing a magnetite solution through water in a sonication bath.
  5. Page 5, line 189-191, we added the reference of Zatsiupa et al.:: “This value is very compatible with the experimental and theoretical results reported by Zatsiupa et al., for the case of Bi25FeO39 at a temperature of 5-950 K and a magnetic field of 0.86 T [19]”.
  6. At the end of secondary heading for 2.1, 2.2, 2.3, 3.3 and 3.5 without full stops, and also for 3.1, 3.2, without full stop.
  7. Page 6 Line 200-202: Increasing the value of μ_eff by 4 times is expected to reduce the value of the magnetic force up to four times [1] to eliminate the suspected target in magnetic hyperthermia application.
  8. Page 6 Line 204-207: Analysis of the distribution of the anisotropic energy barriers of the system, which is closely related to the distribution of nanoparticle size and blocking temperature [13-15], can be through the and  derivative curves to temperature, [],

has been changed to be:

The distribution of the anisotropic energy barriers of the system, which is closely related to the distribution of nanoparticle size and blocking temperature [13-15], can be analyzed through the  and  derivative curves to temperature, [].

  1. Page 6 Line 212: widening has been changed to be broadening.
  2. Page 8 Line 261-263: The decreasing of relaxation time after encapsulation can be correlated with the decrease of medium viscosity, which becomes potential information for biomedical applications [22].
  3. Page 8 Line 266-270: In order to obtain more comprehensive information, it is necessary to conduct further research on the magnetic properties and relaxation time depending on the size of the core Fe3O4. In addition, it is also necessary to be studied the relaxation time depending on the magnetic field. This study is very important to provide information about material properties for biomedical applications that cannot be obtained from in-vivo studies.
  4. We revised the term of “temperature dependent” to be “temperature dependence”
  5. We revised the term of “time dependent” to be “time dependence”
  6. (M-T) has been changed to be M(T)
  7. (M-t) has been changed to be M(t)
  8. We have revised conclusion page 9 Line 272-296 to be:

The magnetic properties of Fe3O4 and its encapsulation with SiO2 (Fe3O4.SiO2) have been successfully investigated by analyzing the () and () curves of the SQUID magnetometer measurements. The () curve for Fe3O4 and Fe3O4.SiO2 materials show superparamagnetic properties with magnetic irreversibility characteristics. After encapsulation, the broadening peak of first derivative of the difference between field cooled and zero field cooled magnetization shifted from 281.6 K to 279 K, which indicates that the Fe3O4.SiO2 material has better superparamagnetic properties than that of the Fe3O4 sample. The values of , , and  increased significantly after encapsulation, which also supported the enhancement of superparamagnetic properties of Fe3O4.SiO2. The value of the effective magnetic moment after encapsulation increased four times from 5.77  to 25.49 . This increment is probably due to the emergence of superparamagnetism from single-domain nanomagnets. Increasing the value of  by 4 times is expected to reduce the value of the magnetic force to eliminate the suspected target in magnetic hyperthermia application.

The () curve for Fe3O4, and Fe3O4.SiO2 shows that the magnetic relaxation time of Fe3O4-SiO2 sample decreases compared to the magnetic relaxation time of Fe3O4. The lower relaxation times are associated to better superparamagnetic properties. This also indicates that the superparamagnetic properties increase with the encapsulation process. The decreasing of relaxation time after encapsulation from  to 3.57 can be correlated with the decrease of medium viscosity, which become potential information for biomedical applications.

In order to obtain more comprehensive information, it is necessary to conduct further research on the magnetic properties and relaxation time depending on the size of the core Fe3O4 to provide information about material properties for biomedical applications that cannot be obtained from in-vivo studies.

  1. We added new reference number 22:

[22] Krishnan, K.M. Biomedical Nanomagnetics: A Spin Through Possibilities in Imaging, Diagnostics, and Therapy. IEEE Trans-actions on Magnetics 2010, 46, 2523-2558.

Reviewer 2 Report

In this article, the authors present the research results of the magnetic properties and relaxation time of Fe3O4 nanoparticles and their encapsulation with silicon dioxide (Fe3O4.SiO2). These properties have been studied by analysing the temperature dependent of magnetization (M(T)) and the time dependent of magnetization (M(t)) using the SQUID magnetometer measurement. The results of the research allowed the authors determine the magnetic parameters and magnetic irreversibility of Fe3O4 and Fe3O4.SiO2 samples. From the results of the M(t) measurement, Authors found that there was a decrease in the magnetic relaxation time after the encapsulation process, which indicated that the distribution of nanoparticle size and anisotropy energy increased. Below I presented some remarks that came to my mind during reading:

  1. In my opinion the Introduction must be improved. Introduction should adequately represent the state of knowledge and clearly specify the purpose and motivation of taking up the topic. The area of research must be introduced with details for unfamiliar readers. The Authors should state what is special, unexpected, or different in their approach. The authors should perform a critical survey of what has been done up to this point in the scientific literature and identify a precise gap in the current state of knowledge that needs to be filled, a gap that is being addressed by their research.
  2. Line 76: Could the authors provide more information on the used nanoparticles (CAS number, etc.)?
  3. Line 84: What were the sonication parameters (amplitude, energy, etc.)?
  4. Conclusions should be worded slightly different. Try to emphasize novelty. Put some quantifications, and comment on the limitations. This is a very common way to write conclusions for a learned academic journal. The conclusions should highlight the novelty and advance in understanding presented in the work. In the Conclusions also, it would be useful to add information on further research of the authors related to the continuation of this research topic.

Author Response

Respond to Reviewer (2)

 Comments and Suggestions for Authors

In this article, the authors present the research results of the magnetic properties and relaxation time of Fe3O4 nanoparticles and their encapsulation with silicon dioxide (Fe3O4.SiO2). These properties have been studied by analysing the temperature dependent of magnetization (M(T)) and the time dependent of magnetization (M(t)) using the SQUID magnetometer measurement. The results of the research allowed the authors determine the magnetic parameters and magnetic irreversibility of Fe3O4 and Fe3O4.SiO2 samples. From the results of the M(t) measurement, Authors found that there was a decrease in the magnetic relaxation time after the encapsulation process, which indicated that the distribution of nanoparticle size and anisotropy energy increased.

Respond: Authors would like to thank for valuable comments and suggestions to increase the quality of our manuscript.

Below I presented some remarks that came to my mind during reading:

Reviewer (1):

  1. In my opinion the Introduction must be improved. Introduction should adequately represent the state of knowledge and clearly specify the purpose and motivation of taking up the topic. The area of research must be introduced with details for unfamiliar readers. The Authors should state what is special, unexpected, or different in their approach. The authors should perform a critical survey of what has been done up to this point in the scientific literature and identify a precise gap in the current state of knowledge that needs to be filled, a gap that is being addressed by their research.

Respond (1):

We have improved the “Introduction”. We have added the gap of research that needs to be filled displaying in page 2 Line 66-71: “Therefore, the determination of the value of  and other magnetic properties in the temperature range of 10-300 K is still needed to be investigated. In addition, the effect of SiO2 encapsulation to Fe3O4 on the effective magnetic moment and relaxation time has not been widely reported. This information is beneficial for its development and application in biomedicine, which is difficult to obtain within in-vivo research.”

Reviewer (2):

  1. Line 76: Could the authors provide more information on the used nanoparticles (CAS number, etc.)?

Respond (2):

We have added the CAS number of precursor with narration page 2 Line 81-83: “The precursors of Fe2+ and Fe3+ cations are ferric chlorid hexahydrate, FeCl3.6H2O (Hoshi Rikagakukikai #15-1140-5-25G-J >99%), and iron (II) chloride tetrahydrate puriss. p.a, FeCl2.4H2O (Hoshi Rikagakukikai # 44939-5G >99%), respectively”.

Reviewer (3):

  1. Line 84: What were the sonication parameters (amplitude, energy, etc.)?

Respond (3):

We have rechecked the sonication parameters and fixed the sonication time and temperature as we did in our experiments. The explanation about sonification time and temperature is displayed on page 2 line 90: “Then the solution was stirred and sonicated at room temperature for 30 minutes”.

We also added detailed sonication information on page 2, line 91:

In this study, the sonication process was carried out simply by using a sonication bath without a sonicator tip. The sonication bath was filled with water, and a beaker glass containing the magnetite solution was put into the sonication bath. The frequency used in the sonication bath is 40 kHz. This ultrasonic wave will be forwarded to a glass beaker containing a magnetite solution through water in a sonication bath.

Reviewer (4):

  1. Conclusions should be worded slightly different. Try to emphasize novelty. Put some quantifications, and comment on the limitations. This is a very common way to write conclusions for a learned academic journal. The conclusions should highlight the novelty and advance in understanding presented in the work. In the Conclusions also, it would be useful to add information on further research of the authors related to the continuation of this research topic.

Respond (4):

We have revised conclusion page 9 Line 272-296 to be:

The magnetic properties of Fe3O4 and its encapsulation with SiO2 (Fe3O4.SiO2) have been successfully investigated by analyzing the M(T) and M(t) curves of the SQUID magnetometer measurements. The () curve for Fe3O4 and Fe3O4.SiO2 materials show superparamagnetic properties with magnetic irreversibility characteristics. After encapsulation, the broadening peak of first derivative of the difference between field cooled and zero field cooled magnetization shifted from 281.6 K to 279 K, which indicates that the Fe3O4.SiO2 material has better superparamagnetic properties than that of the Fe3O4 sample. The values of , , and  increased significantly after encapsulation, which also supported the enhancement of superparamagnetic properties of Fe3O4.SiO2. The value of the effective magnetic moment after encapsulation increased four times from 5.77  to 25.49 . This increment is probably due to the emergence of superparamagnetism from single-domain nanomagnets. Increasing the value of  by 4 times is expected to reduce the value of the magnetic force to eliminate the suspected target in magnetic hyperthermia application.

The M(t) curve for Fe3O4, and Fe3O4.SiO2 shows that the magnetic relaxation time of Fe3O4-SiO2 sample decreases compared to the magnetic relaxation time of Fe3O4. The lower relaxation times are associated to better superparamagnetic properties. This also indicates that the superparamagnetic properties increase with the encapsulation process. The decreasing of relaxation time after encapsulation from  to 3.57 can be correlated with the decrease of medium viscosity, which become potential information for biomedical applications.

In order to obtain more comprehensive information, it is necessary to conduct further research on the magnetic properties and relaxation time depending on the size of the core Fe3O4 to provide information about material properties for biomedical applications that cannot be obtained from in-vivo studies.

List of Changes:

  1. Page 2 Line 50, “The magnetic characteristics” has been changed to be “the magnetic properties”.
  2. We have improved the “Introduction”. We have added the gap of research that needs to be filled displaying in page 2 Line 66-71: “Therefore, the determination of the value of and other magnetic properties in the temperature range of 10-300 K is still needed to be investigated. In addition, the effect of SiO2 encapsulation to Fe3O4 on the effective magnetic moment and relaxation time has not been widely reported. This information is beneficial for its development and application in biomedicine, which is difficult to obtain within in-vivo research.”
  3. Page 2 Line 81-83, we added the explanation about specification of precursor: The precursors of Fe2+ and Fe3+ cations are ferric chlorid hexahydrate, FeCl3.6H2O (Hoshi Rikagakukikai #15-1140-5-25G-J >99%), and iron (II) chloride tetrahydrate puriss. p.a, FeCl2.4H2O (Hoshi Rikagakukikai # 44939-5G >99%), respectively.
  4. Page 2 Line 90-95: Then the solution was stirred and sonicated at room temperature for 30 minutes. In this study, the sonication process was carried out simply by using a sonication bath without a sonicator tip. The sonication bath was filled with water, and a beaker glass containing the magnetite solution was put into the sonication bath. The frequency used in the sonication bath is 40 kHz. This ultrasonic wave will be forwarded to a glass beaker containing a magnetite solution through water in a sonication bath.
  5. Page 5, line 189-191, we added the reference of Zatsiupa et al.: “This value is very compatible with the experimental and theoretical results reported by Zatsiupa et al., for the case of Bi25FeO39 at a temperature of 5-950 K and a magnetic field of 0.86 T [19]”.
  6. At the end of secondary heading for 2.1, 2.2, 2.3, 3.3 and 3.5 without full stops, and also for 3.1, 3.2, without full stop.
  7. Page 6 Line 200-202: Increasing the value of μ_eff by 4 times is expected to reduce the value of the magnetic force up to four times [1] to eliminate the suspected target in magnetic hyperthermia application.
  8. Page 6 Line 204-207: Analysis of the distribution of the anisotropic energy barriers of the system, which is closely related to the distribution of nanoparticle size and blocking temperature [13-15], can be through the and  derivative curves to temperature,

has been changed to be:

The distribution of the anisotropic energy barriers of the system, which is closely related to the distribution of nanoparticle size and blocking temperature [13-15], can be analyzed through the  and  derivative curves to temperature,

  1. Page 6 Line 212: widening has been changed to be broadening.
  2. Page 8 Line 261-263: The decreasing of relaxation time after encapsulation can be correlated with the decrease of medium viscosity, which becomes potential information for biomedical applications [22].
  3. Page 8 Line 266-270: In order to obtain more comprehensive information, it is necessary to conduct further research on the magnetic properties and relaxation time depending on the size of the core Fe3O4. In addition, it is also necessary to be studied the relaxation time depending on the magnetic field. This study is very important to provide information about material properties for biomedical applications that cannot be obtained from in-vivo studies.
  4. We revised the term of “temperature dependent” to be “temperature dependence”
  5. We revised the term of “time dependent” to be “time dependence”
  6. (M-T) has been changed to be M(T)
  7. (M-t) has been changed to be M(t)
  8. We have revised conclusion page 9 Line 272-296 to be:

The magnetic properties of Fe3O4 and its encapsulation with SiO2 (Fe3O4.SiO2) have been successfully investigated by analyzing the M(T) and M(t) curves of the SQUID magnetometer measurements. The () curve for Fe3O4 and Fe3O4.SiO2 materials show superparamagnetic properties with magnetic irreversibility characteristics. After encapsulation, the broadening peak of first derivative of the difference between field cooled and zero field cooled magnetization shifted from 281.6 K to 279 K, which indicates that the Fe3O4.SiO2 material has better superparamagnetic properties than that of the Fe3O4 sample. The values of , , and  increased significantly after encapsulation, which also supported the enhancement of superparamagnetic properties of Fe3O4.SiO2. The value of the effective magnetic moment after encapsulation increased four times from 5.77  to 25.49 . This increment is probably due to the emergence of superparamagnetism from single-domain nanomagnets. Increasing the value of  by 4 times is expected to reduce the value of the magnetic force to eliminate the suspected target in magnetic hyperthermia application.

The M(t) curve for Fe3O4, and Fe3O4.SiO2 shows that the magnetic relaxation time of Fe3O4-SiO2 sample decreases compared to the magnetic relaxation time of Fe3O4. The lower relaxation times are associated to better superparamagnetic properties. This also indicates that the superparamagnetic properties increase with the encapsulation process. The decreasing of relaxation time after encapsulation from  to 3.57 can be correlated with the decrease of medium viscosity, which become potential information for biomedical applications.

In order to obtain more comprehensive information, it is necessary to conduct further research on the magnetic properties and relaxation time depending on the size of the core Fe3O4 to provide information about material properties for biomedical applications that cannot be obtained from in-vivo studies.

  1. We added new reference number 22:

[22] Krishnan, K.M. Biomedical Nanomagnetics: A Spin Through Possibilities in Imaging, Diagnostics, and Therapy. IEEE Trans-actions on Magnetics 2010, 46, 2523-2558.
